# A Comparative Analysis of the Perception of Cancer Patients and Healthcare Providers (Oncology Physicians, Nurses, Social Workers) in Support of Integrated Community-Linked Cancer Plans

**DOI:** 10.3390/ijerph182111517

**Published:** 2021-11-02

**Authors:** Young Ae Kim, Min Gee Choi, E Hwa Yun, So-Youn Jung, Ah Kyung Park, Hye Ri Choi, Yoon Jung Chang

**Affiliations:** 1National Cancer Center, National Cancer Control Institute, Goyang-si 10408, Gyeonggi-do, Korea; elkim7@gmail.com (Y.A.K.); min903@yonsei.ac.kr (M.G.C.); choi.park.hyeri@ncc.re.kr (H.R.C.); 2National Cancer Center, Division of Cancer Registration & Surveillance, Goyang-si 10408, Gyeonggi-do, Korea; ehwayun@ncc.re.kr; 3Cancer Healthcare Research Branch, National Cancer Center, Goyang-si 10408, Gyeonggi-do, Korea; goje1@ncc.re.kr; 4Social Work Team, National Cancer Center, Goyang-si 10408, Gyeonggi-do, Korea; akpark01@ncc.re.kr

**Keywords:** health policy, oncology, quality in health care, public health, primary care

## Abstract

This study aimed to examine the awareness and status of cancer patients and healthcare providers (physicians, nurses and social workers) regarding community linkage, in order to establish a desirable care plan model in a future research project. The survey was conducted via two methods: face-to-face for cancer patients (*n* = 308) and oncology physicians (*n* = 210), and due to COVID-19 circumstances, online for nurses (*n* = 200) and social workers (*n* = 313). As a result, more than 95% of the healthcare providers responded that cancer patients required community-linked services and discharge plans, whereas 50.7% and 79.2% of cancer patients noted the importance of community-linked services and discharge plans, respectively. Social workers, among healthcare providers, showed the most positive experience about connecting patients to community services since 69.7% of them responded as “excellent”. However, as a group, cancer patients considered the necessity of community-linked service as less important, as only 50.7% responded as agreeing it was necessary. The barriers to community linkage were the lack of communication among the different professions of healthcare providers, and the ambiguity in their roles. The findings of this study will inform future community-linked health research, policies and systems for cancer patients. In particular, an in-depth interview with cancer patients will be required to explore their lack of acknowledgment about the necessity of community-linked services. Therefore, this study is expected to contribute to the improvement and supplementation of cancer policies.

## 1. Introduction

The relative survival rate of cancer patients continues to increase due to the expansion of early cancer screening and the improvement of medical technology. In Korea, the relative survival rate of cancer patients between 2013 and 2017 was 70.4%, representing an increase of 27.5% between 1993 and 1995 [1].

This increase in the rate of cancer survival has highlighted the need for improved quality of life after the discharge of cancer patients. In the United States, the 2005 report by the National Academies Press revealed that improved survival rates raised the number of cancer survivors [2]. As of 2016, the United States had about 15 million cancer survivors, most of whom survived more than five years after treatment, and half were older than 70 years [3]. The preceding studies noted the importance of community-linked integrated care as a joint management measure for the long-term treatment of cancer patients who were also struggling with a high burden of comorbidities [3,4,5,6,7]. In addition, studies on community-linked treatment models have shown that appropriate community-linked care and integrated medical services promote the health of cancer survivors [8]. A study on childhood cancer survivors examined the effectiveness of a discharge plan program which supported caregivers in discharge education, home visits, and telephone consultations, and it reported a decrease in the need for physical care and unplanned hospital admissions of patients participating in the discharge plan [9]. 

However, there is currently a lack of an integrated post-care management system for cancer patients in Korea. Many cancer patients visit general hospitals located in the Seoul metropolitan area for better service, which causes the suffering of elderly or non-metropolitan residents from transportation-related problems such as excessive travel time and inconvenience of transportation in the metropolitan area [10]. In this regard, an appropriate approach to community-linked care can contribute to reducing the socioeconomic costs incurred by patients.

As part of the “inclusive country” policy of 2018, the Korean government has actively promoted community care to provide social services to vulnerable people [11]. Likewise, public health centers and local governments have provided various social services and welfare programs for residents following the national policy, but there is a lack of specific services for cancer patients who are discharged from hospital. As part of the government’s policy, the National Cancer Center is conducting a pilot study (for 100 people) that provides certain community-based care services for cancer patients discharged after surgery. Discharged cancer patients participating in the study will receive counseling services regularly over the phone, after returning to the community. The counseling services include social and psychological support as well as physical well-being, in an integrated manner. The services will be provided via linkage with the closest local community center near the patient’s residence. In order to activate such a community-linked care system, the unfulfilled needs of cancer patients must first be identified, in addition to improving the awareness of healthcare providers regarding community-linked care.

The awareness levels of healthcare providers regarding community-linked health and welfare policies have already been examined in some countries. The United Kingdom enacted the Community Care Act in 1990 to reduce patients’ dependence on facility protection nationwide, allowing patients to receive care in their residence [12]. In the United States, the Affordable Care Act implemented a policy that allows patients to live independently in their daily lives [13]. Japan introduced comity-based integrated care in 2005, providing comprehensive medical and psychological care to the elderly within their residences [14]. A study on a community-based cancer patient setup conducted in Canada noted that it is important to identify patients’ unfulfilled needs (financial resources, connectivity services, etc.) for community-based cancer patient management, and improving projects by identifying the role of health providers [15]. In addition to physicians and nurses, it is necessary to provide integrated management through social worker workforce planning for community-based cancer patients [16].

In Korea, research on the awareness levels of healthcare providers regarding community-linked health and welfare policies is relatively insufficient. To the best knowledge of the authors of this paper, there was no prior study that compared awareness of community-linked services for cancer patients among various healthcare providers. Therefore, this survey study aimed to examine the awareness and status of cancer patients and healthcare providers, physicians, nurses and social workers, about community linkage, in order to establish a desirable care plan model in a future research project.

## 2. Materials and Methods

### 2.1. Research Design

From June 2019 to July 2020, the authors surveyed cancer treatment oncology physicians, nurses, social workers, and patients with surgical experience after a cancer diagnosis. The healthcare provider participants were limited to experts with experience in case management, medical treatment, or nursing. The survey was conducted via two methods: face-to-face for cancer patients (*n* = 308) and oncology physicians (*n* = 210), and online for nurses (*n* = 200) and social workers (*n* = 313) due to coronavirus disease (COVID-19) circumstances.

### 2.2. Research Targets and Data Collection

The participants read a description of community-linked cancer care before the survey was conducted. The community-linked cancer care service is a multi-disciplinary patient support team of physicians, nurses and social workers that provides and monitors the medical, social and community-linked information required by patients from hospitalization to discharge.

#### 2.2.1. Oncology Physicians

The survey was limited to “oncology physicians” across the country who worked in hospitals and had direct experience of treating cancer patients. Educated professional researchers surveyed oncology physicians at the KSMO Immune Oncology Forum of the Korean Society of Cancer Science, the WFNOS 2021 Symposium of the Korean Neurological Cancer Society, and the Korean Breast Cancer Radiation Treatment Research Institute. Written surveys were conducted at the National Cancer Center. Accordingly, a total of 210 surveys were completed and used for analysis. In the case of physicians, face-to-face surveys were conducted at cancer-related conferences, and the consent form and survey were provided to physicians who were willing to voluntarily participate.

#### 2.2.2. Social Workers

The survey was limited to “medical social workers” across the country who worked in general hospitals and had experience in case management of cancer patients (or chronic patients) and “local social workers” with certificates of social work and experience in the case management of cancer patients (or chronic patients). Online surveys were conducted in cooperation with the Korean Association of Social Workers and the Korean Association of Medical Social Workers, and a total of 313 surveys were completed and used for analysis. The online survey consisted of a system in which social workers accessing the website participated in the survey through a link, which was marked as banners on each association’s website.

#### 2.2.3. Nurse

The survey was limited to “oncology nurses” across the country who worked in general hospitals and had experience in caring for cancer patients. The online survey was conducted in cooperation with the Korean Nurses Association. The survey was advertised as banners on the websites of a Korean nurse community and a Korean social worker community. Subsequently, participants who clicked and consented voluntarily participated in our survey. A total of 200 surveys were completed and used for analysis.

#### 2.2.4. Cancer Patients

The research focused on patients who had cancer surgery and were discharged from the National Cancer Center (adults over 19 years of age). Written surveys were conducted only by those who were willing to participate at the National Cancer Center, and a total of 308 surveys were completed and used for analysis.

### 2.3. Method of Analysis

To investigate the current status of local community-linked service for cancer patients, a survey was created with reference to relevant domestic and international surveys, which were revised and supplemented through expert advice (provided by a health care professional with more than seven years’ experience in providing care for cancer patients). The questions that investigated the need for community linkage were derived from ‘Public attitudes toward cancer and cancer patients: a national survey in Korea’ [17]; questions regarding the degree of knowledge of community linkage and the experience of community linkage were developed by the researchers. For investigating community-linked experience, additional questions were asked about the services they linked, and the responses were subdivided into four categories: emotional, practical, financial and information, based on advice from Macmillan Cancer Support [18]. In addition, questions about the difficulties of community linkage were added [19,20,21]. The questions asked about the need for community-linked resources for discharged patients, such as meals, traffic, isolation, care, employment, utility bills, housing, and information, referring to the scale of the unfulfilled need screening survey for discharged patients at the National Cancer Center, based on the ‘Social Determinants of Health Tool’ [22]. In addition, the researchers investigated the demographic characteristics of the group of healthcare providers, which included gender, age, agency, work experience, etc., and the demographic characteristics of the group of patients, which included gender, age, diagnosed cancer type, cancer staging, marital status, employment status, insurance type, and monthly household income. Stata 14.0 was used to conduct descriptive statistical analysis of each subject’s demographic characteristics, community connection awareness and connection experience, and connection status.

## 3. Results

### 3.1. The Characteristics of Study Participants

There were more male than female oncology physicians (61% vs. 39%), and the majority of the oncology physicians (*n* = 91, 43.3%) were aged below 40 years [Table 1]. Regarding the type of institution, most oncology physicians (*n* = 147, 70%) worked at the General Hospital. The years of experience also differed, with the majority having an experience of 13 years and above (*n* = 57, 27.1%), followed by 10–13 years (*n* = 56, 26.7%), 5–10 years (*n* = 56, 26.7%), and below 5 years (*n* = 41, 19.5%). Among the nurses, the majority were female (*n* = 187, 93.5%), and the majority of the nurses were aged under 40 years (*n* = 119, 59.5%). Additionally, 96 nurses (48.0%) worked at the General Hospital, and 125 nurses (62.5%) had less than five years of experience. Similarly, among social workers, the majority were women (*n* = 241, *n* = 77%), and the most dominant age group was under 40 years (*n* = 189, 60.4%). Most social workers worked in a social welfare facility (*n* = 135, 43.1%), followed by those who worked in a secondary hospital (*n* = 114, 36.4%). In terms of work experience, most social workers (*n* = 90, 28.8%) had worked for less than five years. In the patient group, women also comprised the majority (*n* = 203, 65.9%), and most patients (*n* = 93, 30.2%) were aged 50–60 years. Regarding the types of cancer, gynecologic cancer was the most common (22.7%), followed by breast cancer (22.4%), colorectal cancer (21.4%), gastric cancer (17.5%), and lung cancer (15.9%). In cancer staging, the third stage was the most common (*n* = 94, 30.5%).

### 3.2. Recognition of Community Linked Service

More than 95% of healthcare providers (oncology physicians, nurses, and social workers) said they needed community links for cancer patients [Table 2]. Regarding recognition of community-linked service by patients, 50.7% of patients indicated that they required community links whilst the others responded that the community links were not needed. In contrast, 79.2% of cancer patients and more than 95% of the healthcare providers noted that a discharge plan was necessary. Most social workers (*n* = 231, 73.8%) noted that they had heard about community-linked service for cancer patients, followed by nurses (*n* = 118, 59.0%) and oncology physicians (*n* = 92, 43.8%). The highest experience rate of linking patients to the community was reported by social workers (*n* = 218, 69.7%), followed by oncology physicians (*n* = 59, 28.1%), nurses (*n* = 38, 19.0%), and patients (*n* = 49, 15.9%).

### 3.3. Community Connection Status if Linkage Is Experienced

According to Table 3, community connection status also differed across the three healthcare provider groups. The main service category linked by each group was practical service by 66.7% of oncology physicians, emotional service by 42.3% of nurses, and financial service by 69.3% of social workers. Regarding difficulties in undertaking community-linkage services, physicians found that the lack of communication between tumor specialists and local service providers was the main challenge (26.1%), while nurses and social workers cited ambiguity of their roles as the main problem (14.4% and 36.7%, respectively).

## 4. Discussion

In this study, it was found that more than 95% of healthcare providers (oncology physicians, nurses, and social workers) acknowledged the need for community-linked care, and the availability of a discharge plan for cancer patients. Cancer patients also recognized community-linked service, with 50.7% and 79.2% of the patients acknowledging the importance of community connection and a discharge plan, respectively. In the future, it will be necessary to investigate the reasons for the negative responses by patients regarding the importance of community connection and experience in community-linked care. Among the three groups of healthcare providers, social workers accounted for the largest group linking patients to the community. This study has shown that only a few cancer patients (15.9%) have experienced community-linked service. In the future, it will be necessary to investigate whether the negative responses by patients about the importance of community connection and experience in community-linked care are due to a lack of knowledge about community-linked care.

Another finding was the different roles of various healthcare providers in community linkage services. While physicians mainly linked actual services, nurses and social workers mainly linked emotional services and financial services. Based on these results, it is necessary to provide professional training on psychological counseling methods to nurses and social workers. A prior study on the professional role of social workers in cancer care showed that this profession is necessary for the psycho–social treatment of cancer patients and requires professional training in counseling [23].

The main barriers to community linkage differed across healthcare providers, with physicians citing lack of communication, and nurses and social workers decrying role ambiguity. The findings are somewhat in line with the findings of the preceding study. In Korea, a study of attitudes toward linkage of clinical care and community-based health care found that more than 50 percent of the interviewed 635 social and healthcare workers noted that medical institutions and social services should be closely linked without duplicating their duties [24]. Therefore, the successful implementation of community-linked services will require effective communication among healthcare providers. Additionally, the government should provide guidelines on the role of each healthcare provider to prevent role ambiguity and ensure effective communication. These findings will help develop desirable policies to facilitate the formulation and implementation of community-linked services. The strength of this study is the comparison and analysis of the perception of community-linked health and welfare systems of cancer patients and three groups of healthcare providers. This study is meaningful as it provides new knowledge about community linkage recognition of cancer patients and can be used to establish related policies and systems in the future. Despite this research value, this study has some limitations. First, since this study was conducted as an exploratory study to investigate only the overall perception of community linkage awareness, more in-depth research is needed. Second, face-to-face surveys were initially planned, but due to the COVID-19 pandemic, they were changed to online surveys, and the participation rate was lower than expected. Third, as the implementation of community-based service in South Korea has a short history, it was difficult to access the results of community-based cancer service to compare with the results of this study. Therefore, it will be necessary to undertake another study with a higher participation rate, and follow-up research to improve community linkage service systems and policies.

The authors look forward to using these findings to establish a desirable direction for future community-linked care plans for cancer patients.

## 5. Conclusions

The lack of effective communication and ambiguity of roles among healthcare providers as obstacles to community-linked service for cancer patients have also been identified in prior studies. First, in Hong Kong, the absence of a standardized discharge plan program and lack of communication and coordination among health care providers hindered proper discharge planning [25]. Second, an in-depth analysis of the role of medical social workers in a multidisciplinary team for cancer care in the United States revealed that poor communication and interpersonal relationships with other professions, such as physicians and nurses, prevented social workers from adapting to the clinical care environment [26]. These findings are in line with the findings in this paper that healthcare providers lack clarity and professional communication about their roles. When planning a community-based discharge plan program for cancer patients in the future, it will be necessary to facilitate communication between occupations through a multidisciplinary approach, and establish a manual for each occupation.

According to the operational plan of the National Cancer Survivor Support Center [27], the unfulfilled needs of cancer patients should be identified through in-depth evaluation, and personnel with experience and expertise should be recruited to provide integrated support to cancer survivors. It is believed that the results of this study in the profession of managing cancer patients in Korea can contribute to the improvement of desirable policies.

## Figures and Tables

**Table 1 ijerph-18-11517-t001:** The characteristics of study participants.

Characteristics	Oncology Physician	Nurse	Social Worker	Cancer Patient
Number of observations	210	200	313	308
Gender N (%)				
Male	128 (61.0)	13 (6.5)	72 (23.0)	105 (34.1)
Female	82 (39.0)	187 (93.5)	241 (77.0)	203 (65.9)
Age group N (%)				
~40	91 (43.3)	119 (59.5)	189 (60.4)	10 (3.2)
40~50	85 (40.5)	54 (27.0)	86 (27.5)	56 (18.2)
50~60	31 (14.8)	27 (13.5)	33 (10.5)	93 (30.2)
60~70	3 (1.4)	0 (0.0)	4 (1.3)	82 (26.6)
70~	0 (0.0)	0 (0.0)	1 (0.3)	67 (21.8)
Type of institution N (%)				
General hospital	147 (70.0)	96 (48.0)	60 (19.2)	
Secondary hospital	59 (28.1)	83 (41.5)	114 (36.4)	
General practitioner	2 (1.0)	0 (0.0)	2 (0.6)	
Social welfare facility	0 (0.0)	7 (3.5)	135 (43.1)	
Research institute	2 (1.0)	14 (7.0)	2 (0.6)	
Years of working N (%)				
~5	41 (19.5)	125 (62.5)	90 (28.8)	
5~10	56 (26.7)	39 (19.5)	79 (25.2)	
10~13	56 (26.7)	20 (10.0)	72 (23.0)	
13~	57 (27.1)	16 (8.0)	72 (23.0)	
Type of cancer N (%)				
Colorectal cancer				66 (21.4)
Gynecologic cancer				70 (22.7)
Gastric cancer				54 (17.5)
Breast cancer				69 (22.4)
Lung cancer				49 (15.9)
Cancer staging N (%)				
0				4 (1.3)
1				79 (25.7)
2				68 (22.1)
3				94 (30.5)
4				63 (20.5)

**Table 2 ijerph-18-11517-t002:** Recognition of community-linked care.

Characteristics	Oncology Physician	Nurse	Social Worker	Cancer Patient
Number of observations	210	200	313	308
Do you think community connection is required for cancer patients? N (%)				
Yes	207 (98.6)	199 (99.5)	308 (98.4)	156 (50.7)
No	3 (1.4)	1 (0.5)	5 (1.6)	152 (49.4)
Do you think a discharge plan is required for cancer patients? N (%)				
Yes	203 (96.7)	199 (99.5)	308 (98.4)	244 (79.2)
NO	7 (3.3)	1 (0.5)	5 (1.6)	64 (20.8)
Have you ever heard of community-linked care for cancer patients? N (%)				
Yes	92 (43.8)	118 (59.0)	231 (73.8)	
NO	118 (56.2)	82 (41.0)	82 (26.2)	
Do you have experience in community-linked care for patients? N (%)				
Yes	59 (28.1)	38 (19.0)	218 (69.7)	49 (15.9)
NO	151 (71.9)	162 (81.0)	95 (30.4)	259 (84.1)

**Table 3 ijerph-18-11517-t003:** Community connection status if linkage is experienced.

Characteristics	Total	Oncology Physician	Nurse	Social Worker
Provided service N (%)				
Emotional	101 (14.3)	16 (23.2)	47 (42.3)	38 (17.4)
Practical	110 (15.5)	46 (66.7)	37 (33.3)	27 (12.4)
Financial	171 (24.2)	5 (7.2)	15 (13.5)	151 (69.3)
Information and advice	16 (2.3)	2 (2.9)	12 (10.8)	2 (0.9)
Difficulty in connecting with local service N (%)				
Role ambiguity	106 (15.0)	10 (14.5)	16 (14.4)	80 (36.7)
Lack of communication between oncologists and service providers	64 (9.0)	18 (26.1)	4 (3.6)	42 (19.3)
Lack of trust of patients	47 (6.6)	11 (15.9)	0 (0.0)	36 (16.5)
Lack of information	29 (4.1)	7 (10.1)	3 (2.7)	19 (8.7)
Lack of financial support	21 (3.0)	8 (11.6)	0 (0.0)	13 (6.0)
Workload and the pressure of time	12 (1.7)	1 (1.4)	0 (0.0)	11 (5.0)
Lack of ability of primary care	4 (0.6)	3 (4.3)	0 (0.0)	1 (0.5)
Lack of patient information	1 (0.1)	1 (1.4)	0 (0.0)	0 (0.0)
Lack of connected institutions, lack of resources, absence of cancer support information system	16 (2.3)	0 (0.0)	0 (0.0)	16 (7.3)
No answer	10 (1.4)	10 (14.5)	0 (0.0)	0 (0.0)

## Data Availability

Data are available upon reasonable request by emailing eunicemd@ncc.re.kr.

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
