# Peer review of "A Comparative Analysis of the Perception of Cancer Patients and Healthcare Providers (Oncology Physicians, Nurses, Social Workers) in Support of Integrated Community-Linked Cancer Plans"

_ijerph, 2021, doi:10.3390/ijerph182111517_

Round 1

Reviewer 1 Report

Recommendations to authors:

Improve language in writing, take care both in phrasing and in the use of non-repetitive words.

Up-to-date citation.

There are results that have not been commented on in the discussion, connect both sections.

Author Response

Response to Reviewer 1

Comments and Suggestions for Authors:

  1. Improve language in writing, take care both in phrasing and in the use of non-repetitive words.

We agree with the reviewer that improving language in writing in this study is important. Based on the reviewer feedback, we tried our best to paraphrased sentences and finished proofreading.

  1. Up-to-date citation.

Thank you for giving this important suggestion. Following the comments, we have added up-to-date literature related to cancer-related community linked care.

Introduction (page 2, line 64-70)

The study on community-based cancer patient setup conducted in Canada noted that identifying patients' unfulfilled needs (financial resources, connectivity services, etc.) for community-based cancer patient management and improving projects by identifying the role of healthcare providers. [12] In addition to physicians and nurses, it is necessary to provide integrated management through social worker workforce planning for such community-based cancer patients.[13]”

  1. There are results that have not been commented on in the discussion, connect both sections.

Thank you for giving this important suggestion. We have added results in discussion. Also, we added these lessons to conclusion.

Discussion (page 7, line 213-215)

In a prior study on the professional role of social workers in cancer care shows that this profession is necessary for the psycho-social treatment of cancer patients and requires professional training in counseling [23].

Discussion (page 7, line 218-222)

“The findings are somewhat in line with the findings of the preceding study. In Korea, a study of attitudes toward linkage of clinical care and community-based health care found that more than 50 percent of the interviewed 635 social and healthcare workers noted that medical institutions and social services should be closely linked without duplicating their duties [24].”

Conclusions (page 7, line 253-257)

“These findings are in line with the findings in this paper that healthcare providers lack ambiguity and professional communication about their roles. When planning a community-based discharge plan program for cancer patients in the future, it is necessary to facilitate communication between occupations through a multidisciplinary approach and establish a manual for each occupation”

Conclusions (page 8, line 261-263)

It is believed that the results of this study in the profession of managing cancer patients in Korea can contribute to the improvement of desirable policies.”

Reviewer 2 Report

Thank you for the opportunity to review this descriptive study considering interest in community-linked care across cancer patients, and healthcare providers. The results provide clear guidance on next steps that could be taken to increase use of community-linked care in Korea. The assessment across types of healthcare providers was also a strength. Comments and suggestions for clarifications follow. 

Minor - Some English language issues here and there reduced clarity. For example, the last sentence of the Abstract starts, "These results provide in-depth interviews in the future". It is unclear if the authors are saying that there will be in depth interviews in the future or that the results of this study, which was in-depth but not a structured interview, can help with future policies.  

Introduction: 
-Lines 63-79: Is there any literature that speaks more directly to awareness among healthcare providers about community-based health and welfare programs that is specific to cancer? The introduction up to this point is very focused on cancer, and the shift away from that in this paragraph was surprising. Especially because the authors do not state that this hasn't been considered for cancer before diving into other disease settings/contexts.

Methods:
-Any sense of the response rates for these surveys? For example, the social worker survey was distributed by email. Are the authors able to report how many invitation emails were sent?

- The analysis component of the methods section needs more elaboration. It appears this was a descriptive study, so a statement that descriptive statistics should be included, and a statement about what was estimated (means? Percentages?). Did the authors consider looking at any associations of interest for their primary outcomes? The Discussion section says differences by patient characteristics were considered but an analytic plan looking at those differences is not described in the Methods and no Results are presented with this information. 

- Concerns about the item assessing interest in community-linked care. The question assessing linked care in Table 2 is worded, “Do you think community connection is required for cancer patients?” As written, it is not clear how this statement is interpreted as community LINKED to care. This statement could also be interpreted as a general community connection being useful for cancer patients. Did the original item specifically mention community-located services or programs? Given the potential ambiguity in the question, it might be less surprising to have found relatively low endorsement among cancer patients themselves. Did the authors consider this potential limitation?

Discussion:
Lines 193-196. This statement seems a little too specific, by providing only one potential explanation of why patients may experience barriers to community-linked care. I would suggest rephrasing to something like, "In the future it will be necessary to investigate the reasons for the negative responses by patients about the importance of community connection and experience in community-linked care".

Lines 204-206. These results are not presented in the manuscript.

The information presented in lines 224-231 would be better suited in the main section of the Discussion section to add more depth to the findings. Currently, the first paragraph of the Discussion section provides an overall summary of all findings but lack comparison of the current study results to previous research. Further, for barriers related to communication, are the authors aware of any clinical interventions designed to assist in communication? How might these fit within current cancer care in Korea?

Author Response

Response to Reviewer 2

Comments and Suggestions for Authors:

  1. Minor - Some English language issues here and there reduced clarity. For example, the last sentence of the Abstract starts, "These results provide in-depth interviews in the future". It is unclear if the authors are saying that there will be in depth interviews in the future or that the results of this study, which was in-depth but not a structured interview, can help with future policies.  

Thank you for your meticulous and careful review. Regarding the English language issues, we made some changes in order for the clarifying and indicating the objectives of this paper in the Abstract.

Abstract (page 1, line 13-15, 29-30)

“This study aims to examine the awareness and status of cancer patients and healthcare providers (physicians, nurses and social workers) about community linkage in order for establishing a desirable care plan model in the future research project.”

“Therefore, this study is expected to contribute to the improvement and supplement of cancer policies.”

  1. Introduction: 
    - Lines 63-79: Is there any literature that speaks more directly to awareness among healthcare providers about community-based health and welfare programs that is specific to cancer? The introduction up to this point is very focused on cancer, and the shift away from that in this paragraph was surprising. Especially because the authors do not state that this hasn't been considered for cancer before diving into other disease settings/contexts.

Following your feedback, we have added literature related to cancer-related community linked care.

Introduction (page 2, line 74-80)

The study on community-based cancer patient setup conducted in Canada noted that identifying patients' unfulfilled needs (financial resources, connectivity services, etc.) for community-based cancer patient management and improving projects by identifying the role of health care providers. [12] In addition to physicians and nurses, it is necessary to provide integrated management through social worker workforce planning for such community-based cancer patients.[13]”

  1. Methods:
    - Any sense of the response rates for these surveys? For example, the social worker survey was distributed by email. Are the authors able to report how many invitation emails were sent?

- The analysis component of the methods section needs more elaboration. It appears this was a descriptive study, so a statement that descriptive statistics should be included, and a statement about what was estimated (means? Percentages?). Did the authors consider looking at any associations of interest for their primary outcomes? The Discussion section says differences by patient characteristics were considered but an analytic plan looking at those differences is not described in the Methods and no Results are presented with this information. 

- Concerns about the item assessing interest in community-linked care. The question assessing linked care in Table 2 is worded, “Do you think community connection is required for cancer patients?” As written, it is not clear how this statement is interpreted as community LINKED to care. This statement could also be interpreted as a general community connection being useful for cancer patients. Did the original item specifically mention community-located services or programs? Given the potential ambiguity in the question, it might be less surprising to have found relatively low endorsement among cancer patients themselves. Did the authors consider this potential limitation?

We agree with the opinion shared. As you gave us feedback, we added an explanation to clarify the investigation method. Also, we provided the survey participants with a description of community-linked cancer care before the survey was conducted. These processes are more clearly described in the method section.

Materials and Methods (page 2, line 99-103)

The participants had a chance to read a description of community-linked cancer care before the survey was conducted. The Community-linked cancer care service is a multidisciplinary patient support team of physicians, nurses and social workers that provides and monitors the medical, social and community-linked information required by patients from hospitalization to discharge.”

Materials and Methods (page 3, line 120-122)

“The online survey consisted of a system in which social workers accessing the website participated in the survey through the link, marked as banners on each association's website.”

Materials and Methods (page 4, line 152-154)

“We used Stata 14.0 to conduct descriptive statistical analysis of each subject's demographic characteristics, community connection awareness and connection experience, and connection status”

  1. Discussion:
    - Lines 193-196. This statement seems a little too specific, by providing only one potential explanation of why patients may experience barriers to community-linked care. I would suggest rephrasing to something like, "In the future it will be necessary to investigate the reasons for the negative responses by patients about the importance of community connection and experience in community-linked care".

Thank you for your detailed suggestion. Following your comment, we paraphrased the part in the paper.

Discussion (page 6, line 206-208)

In the future, it will be necessary to investigate the reasons for the negative responses by patients about the importance of community connection and experience in community-linked care

- Lines 204-206. These results are not presented in the manuscript.

Thank you for your meticulous and careful review. We deleted this part following your comment.

- The information presented in lines 224-231 would be better suited in the main section of the Discussion section to add more depth to the findings. Currently, the first paragraph of the Discussion section provides an overall summary of all findings but lack comparison of the current study results to previous research. Further, for barriers related to communication, are the authors aware of any clinical interventions designed to assist in communication? How might these fit within current cancer care in Korea?

Following your comment, we added the results of the first paragraph and the comparison with the current study, and we moved the communication-related part to the discussion section.

Conclusions (page 7, line 253-257)

“These findings are in line with the findings in this paper that healthcare providers lack ambiguity and professional communication about their roles. When planning a community-based discharge plan program for cancer patients in the future, it is necessary to facilitate communication between occupations through a multidisciplinary approach and establish a manual for each occupation”

Reviewer 3 Report

This is a descriptive study on the community-based cancer care programs in Korea.  Information provided is very helpful to understand how community-based cancer care plans are implemented and perceived by cancer patients and selected providers.  The paper has some merits in terms of its relevance to post-hospital care that is integrated to community services delivered by providers.  The paper could be strengthened in the following areas:

  1. Clarification the terms such as community-based care vs. community-linked care:  Are they the same plans or programs delivered in Korea?
  2. Illustration of services provided:  Are the services offered at home or a community service setting?  What do they do for cancer patients?
  3. Description on the community-based interventions provided:  Are their specific behavioral or family interventions at the community-based care setting?
  4. Evaluation of the plans or programs: Are there any specific program  evaluation (e.g., outcomes assessment) done for the community-based cancer program?
  5. Lesson: What can be learned from this report?  Are there specific principles or strategies in the delivery of community-based cancer care can be shared with researchers in other nations?

Author Response

Response to Reviewer 3

Comments and Suggestions for Authors:

  1. Clarification the terms such as community-based care vs. community-linked care:  Are they the same plans or programs delivered in Korea?

Thank you for your meticulous and careful review. Following your comment, we changed the terms to “community-linked care” for coherence and consistency. Apparently, the terms, community-based care and community-linked care were not clearly distinguished in Korea due to the short history of Korean community care. Thanks again for the valuable and important notion.

2.

- Illustration of services provided:  Are the services offered at home or a community service setting?  What do they do for cancer patients?

- Description on the community-based interventions provided:  Are their specific behavioral or family interventions at the community-based care setting?

We agree with the opinion shared. Following the comment, we have added specifically what services is offered for community-based setting.

Introduction (page 2, line 63-70)

“As part of the government's policy, the National Cancer Center is conducting a pilot study (for 100 people) that provides certain community-based care services for cancer patients discharging after surgery. Patients who participate in the study regularly receive counseling services over the phone after returning to the community after discharge. In addition to health counseling, counseling will be provided in an integrated manner, including social and psychological counseling, and will be provided in connection with the local community institutions where patients reside.”

Materials and Methods (page 2, line 99-103)

The participants had a chance to read a description of community-linked cancer care before the survey was conducted. The Community-linked cancer care service is a multidisciplinary patient support team of physicians, nurses and social workers that provides and monitors the medical, social and community-linked information required by patients from hospitalization to discharge.”

  1. Evaluation of the plans or programs: Are there any specific program evaluation (e.g., outcomes assessment) done for the community-based cancer program?

We regret to inform you that Korean community-based care has a very short history which was implemented in 2019. Therefore, it was not possible to find and access any program evaluations yet. Nonetheless, we do appreciate this comment since it gave us a valuable notion about the limitation of this study.

  1. Lesson: What can be learned from this report?  Are there specific principles or strategies in the delivery of community-based cancer care can be shared with researchers in other nations?

Thank you for providing detailed suggestions. It is believed that the results of this study in the profession of managing cancer patients in Korea can contribute to the improvement of desirable policies (e.g., compared to the results of countries that provided community-based cancer patient care earlier). we added these lessons to conclusion.

Discussion (page 6, line 206-208)

In the future, it will be necessary to investigate the reasons for the negative responses by patients about the importance of community connection and experience in community-linked care

Conclusions (page 7, line 253-257)

“These findings are in line with the findings in this paper that healthcare providers lack ambiguity and professional communication about their roles. When planning a community-based discharge plan program for cancer patients in the future, it is necessary to facilitate communication between occupations through a multidisciplinary approach and establish a manual for each occupation”

Conclusions (page 7, line 261-263)

It is believed that the results of this study in the profession of managing cancer patients in Korea can contribute to the improvement of desirable policies.”

Round 2

Reviewer 2 Report

This revised manuscript was moderately responsive to previous comments, however, I have remaining concerns. 

-    A major remaining concern is related to response rates. I previously asked the authors if any information is available regarding response rates, and I did not receive a response to this inquiry. For example, the nurse survey was sent out via email distribution, do the authors know how many emails were sent? That would provide at least some estimate of the representativeness of the sample. Similarly for the patient surveys, do the authors know how many cancer patients were approached who declined? Or, how many cancer patients would have been discharged from care and eligible to participate during the time window when patient surveys were collected? As is there is no way to determine the generalizability of the findings.  

-    Second remaining concern pertains to the introduction. My previous review asked, “ Is there any literature that speaks more directly to awareness among healthcare providers about community-based health and welfare programs that is specific to cancer?” The authors revised the introduction section by removing all previous discussion of the literature and presenting on only one study, but still state, “The awareness levels of healthcare providers regarding community- linked health and welfare policies have already been examined in some countries”, which implies this topic has been considered in more than one peer-reviewed study. I also highly doubt that there has really only been one previous publication on this topic in the cancer domain. 

-    Further, the statement, “The need for integration of providers through comprehensive treatment models was also emphasized [14].” No longer makes sense without the description of the study that it is referencing, which had been the sentence before this one in the previous version.

-    Lines 206-209. The authors state in their response that these lines were modified to, “In the future, it will be necessary to investigate the reasons for the negative responses by patients about the importance of community connection and experience in community-linked care”, but the lines were not changed. 

-    I previously pointed out that lines 226-228, “Moreover, our study showed that the perspective of community-linked care differed depending on the patient's gender, type of cancer, cancer staging, and employment status.” Either needed to be removed or the data showing these results needed to be added. The authors response states that those lines were deleted, but they remain without the addition of any tables. 

-    Minor - Lines 99-103 would be more appropriate in the “Method of analysis” section since that is where the survey items are described. 

-    Minor – continued English language issues, particularly in new sections, left me confused at some points. In particular, a frequent changing of tenses from present to past tense.

Author Response

Response to Reviewer 2

Comments and Suggestions for Authors:

  1. A major remaining concern is related to response rates. I previously asked the authors if any information is available regarding response rates, and I did not receive a response to this inquiry. For example, the nurse survey was sent out via email distribution, do the authors know how many emails were sent? That would provide at least some estimate of the representativeness of the sample. Similarly for the patient surveys, do the authors know how many cancer patients were approached who declined? Or, how many cancer patients would have been discharged from care and eligible to participate during the time window when patient surveys were collected? As is there is no way to determine the generalizability of the findings.  

Thank you for informing us about the concern in very specific. However, unfortunately, the response rates could not be measured in our study due to the type of survey we conducted. First, the survey for nurses and social workers recruited participants via online advertisements. Our survey was advertised as banners on the websites of a Korean nurse community and the Korean social worker community. Subsequently, participants who clicked and consented voluntarily participated in our survey. In terms of physicians, face-to-face recruitment for the survey was conducted at cancer-related conferences. Consent form and the survey were only provided to physicians who are willing to participate voluntarily. Likewise, patient participants who wished to participate were recruited for the surveys. Therefore, the response rate could not be measured in this research since the survey recruited participants who are voluntarily willing only. The detailed explanation was added to the materials and methods section as the lines below.

(Materials and Methods, page 3, line 116-118)

In the case of physicians, face-to-face surveys were conducted at cancer-related conferences, and consent form and survey was provided to physicians who are willing to participate voluntarily.”

(Materials and Methods, page 3, line 132-135)

“The online survey was conducted in cooperation with the Korean Nurses Association. Our survey was advertised as banners on the websites of a Korean nurse community and the Korean social worker community. Subsequently, participants who clicked and consented voluntarily participated in our survey.”

(Materials and Methods, page 3, line 139-140)

“Written surveys were conducted only on those who were willing to participate at the National Cancer Center”

  1. Second remaining concern pertains to the introduction. My previous review asked, “ Is there any literature that speaks more directly to awareness among healthcare providers about community-based health and welfare programs that is specific to cancer?” The authors revised the introduction section by removing all previous discussion of the literature and presenting on only one study, but still state, “The awareness levels of healthcare providers regarding community- linked health and welfare policies have already been examined in some countries”, which implies this topic has been considered in more than one peer-reviewed study. I also highly doubt that there has really only been one previous publication on this topic in the cancer domain. 

We agreed with this concern regarding one of the paragraphs in the introduction. Therefore, we added several studies of community- linked health and welfare policies conducted in different countries.

(Introduction, page 2, line 75-81)

The UK enacted the Community Care Act in 1990 to reduce patients' dependence on facility protection nationwide, allowing patients to receive care in their residence[12]. In the case of the United States, the Affordable Care Act implemented a policy that allows patients to live independently of their daily lives[13]. Japan introduced comity-based integrated care in 2005, providing comprehensive medical and psychological care to the elderly within their residences[14].

  1.  Further, the statement, “The need for integration of providers through comprehensive treatment models was also emphasized [14].” No longer makes sense without the description of the study that it is referencing, which had been the sentence before this one in the previous version.

We also agreed here so deleted the sentence.

  1. Lines 206-209. The authors state in their response that these lines were modified to, “In the future, it will be necessary to investigate the reasons for the negative responses by patients about the importance of community connection and experience in community-linked care”, but the lines were not changed. 

We must apologize for this error and are grateful for reminding of. We revised the sentence following your comment.

(Discussion, page 7, line 214-216)

“In the future, it will be necessary to investigate the reasons for the negative responses by patients about the importance of community connection and experience in community-linked care.”

  1. I previously pointed out that lines 226-228, “Moreover, our study showed that the perspective of community-linked care differed depending on the patient's gender, type of cancer, cancer staging, and employment status.” Either needed to be removed or the data showing these results needed to be added. The authors response states that those lines were deleted, but they remain without the addition of any tables. 

We did agree with the previous comment about the sentence. We absolutely admitted that the sentence should have been removed at the previous version. We thank for your generosity of reminding of again.

  1. Minor - Lines 99-103 would be more appropriate in the “Method of analysis” section since that is where the survey items are described. 

We definitely agreed with this. So, the sentences were moved to the "Method of analysis" section.

(Materials and Methods, page 3, line 104-108)

The participants had a chance to read a description of community-linked cancer care before the survey was conducted. The Community-linked cancer care service is a multi-disciplinary patient support team of physicians, nurses and social workers that provides and monitors the medical, social and community-linked information required by patients from hospitalization to discharge.

  1. Minor – continued English language issues, particularly in new sections, left me confused at some points. In particular, a frequent changing of tenses from present to past tense.

Thank you for pointing out the issue. We carefully checked tenses throughout the all sentences.
